

# Conditions during adulthood affect cohort-specific reproductive success in an Arctic-nesting goose population

Mitch D. Weegman[1,2], Stuart Bearhop[1], Geoff M. Hilton[2], Alyn Walsh[3] and Anthony David Fox[4]

[1] Centre for Ecology and Conservation, University of Exeter, Cornwall Campus, Penryn, Cornwall, United Kingdom
[2] Wildfowl & Wetlands Trust, Slimbridge, Gloucestershire, United Kingdom
[3] Wexford Wildfowl Reserve, National Parks and Wildlife Service, Wexford, Ireland
[4] Department of Bioscience, Aarhus University, Rønde, Denmark

Corresponding author
Mitch D. Weegman,
weegm009@umn.edu

## ABSTRACT

Variation in fitness between individuals in populations may be attributed to differing environmental conditions experienced among birth (or hatch) years (i.e., between cohorts). In this study, we tested whether cohort fitness could also be explained by environmental conditions experienced in years post-hatch, using 736 lifelong resighting histories of Greenland white-fronted geese (*Anser albifrons flavirostris*) marked in their first winter. Specifically, we tested whether variation in age at first successful reproduction, the size of the first successful brood and the proportion of successful breeders by cohort was explained by environmental conditions experienced on breeding areas in west Greenland during hatch year, those in adulthood prior to successful reproduction and those in the year of successful reproduction, using North Atlantic Oscillation indices as proxies for environmental conditions during these periods. Fifty-nine (8%) of all marked birds reproduced successfully (i.e., were observed on wintering areas with young) only once in their lifetime and 15 (2%) reproduced successfully twice or thrice. Variation in age at first successful reproduction was explained by the environmental conditions experienced during adulthood in the years prior to successful reproduction. Birds bred earliest (mean age 4) when environmental conditions were 'good' prior to the year of successful reproduction. Conversely, birds successfully reproduced at older ages (mean age 7) if they experienced adverse conditions prior to the year of successful reproduction. Hatch year conditions and an interaction between those experienced prior to and during the year of successful reproduction explained less (marginally significant) variation in age at first successful reproduction. Environmental conditions did not explain variation in the size of the first successful brood or the proportion of successful breeders. These findings show that conditions during adulthood prior to the year of successful reproduction are most important in determining the age at first successful reproduction in Greenland white-fronted geese. Very few birds bred successfully at all (most only once), which suggests that May environmental conditions on breeding areas have cohort effects that influence lifetime (and not just annual) reproductive success.

## INTRODUCTION

Individual variation in fitness is a feature of vertebrate populations (*Gaillard et al., 2000*), some of which results from annual variation in conditions experienced during early life (*Sæther, 1997*), giving rise to 'cohort effects' (*Lindström, 1999*). Cohort effects are well documented in birds (*Van der Jeugd & Larsson, 1998*; *Krüger & Lindström, 2001*; *Reid et al., 2003*) and mammals (*Rose, Clutton-Brock & Guinness, 1998*; *Coltman et al., 1999*; *Descamps et al., 2008*), where subsequent fitness has been linked to birth year conditions via life history traits. For example, Soay sheep (*Ovis aries*) born after warm, wet winters produced more offspring as adults than those born after cold, dry winters (*Forchhammer et al., 2001*). In some birds, juvenile survival, probability of recruitment into the breeding population and breeding longevity were positively correlated with the quality of a cohort's natal environment (*Reid et al., 2003*). Inter-cohort variation in life history traits can help to explain individual performance in relation to conditions experienced by individuals born in the same year. For instance, individuals experiencing 'good' early life conditions may exhibit enhanced fitness compared to those exposed to 'poor' early life conditions, a facet of the so-called 'silver spoon' effect (*Grafen, 1988*; *Cooke, Findlay & Rockwell, 1984*). Nevertheless, prevailing conditions encountered in later life (e.g., population density or weather during the breeding period) will also likely contribute to variation in cohort-specific life history traits because different cohorts experience different conditions during their potential breeding lifespan (*Reid et al., 2003*; *Thessing & Ekman, 1994*; *Reed et al., 2003*). Hence, favorable birth year effects may be offset if cohorts experience adverse conditions during subsequent breeding years. For example, in North American red squirrels (*Tamiasciurus hudsonicus*), silver spoon effects were diluted in cohorts that experienced lower food availability as adults (*Descamps et al., 2008*).

Breeding year conditions may be highly variable, particularly in Arctic regions (*Martin & Wiebe, 2004*). For instance, breeding success in dark-bellied brent geese (*Branta bernicla bernicla)* is mainly dependent on lemming abundance (when predation pressure on geese is reduced because abundant lemmings provide alternative food sources for predators) and the onset of spring at the Arctic nesting grounds (*Nolet et al., 2013*). Recent cohorts have been exposed to a series of summers with low lemming abundance, so reproductive success and population size have declined (*Nolet et al., 2013*). Yet not all individuals breed in a given year, even during favorable breeding conditions (*Sedinger et al., 2008*). Whereas the highest quality individuals may always exploit the first opportunity to breed, lesser quality individuals may require several optimal years to gain condition before breeding, perhaps influenced by conditions experienced from one season to the next (termed 'carry-over effects'; *Inger et al., 2010*; *Harrison et al., 2011*). Carry-over effects may also affect fitness in cohorts of migrant birds, since pre-nesting body condition (which may be influenced by events extending back to previous winter conditions) was correlated with reproductive output and survival at the individual level (*Ebbinge & Spaans, 1995*; *Baker et al., 2004*). Hence, the cumulative effects of prevailing conditions experienced by a cohort from their collective maturity through to the point at which they successfully breed may have profound influence on the variation in age of first breeding among individuals in a given

cohort. Understanding the degree to which these effects influence cohort variation in life history traits is therefore paramount in determining the relative importance of hatch year conditions, those experienced in adulthood prior to successful reproduction and those experienced in the year of successful reproduction.

Here, we used a 21-year dataset of repeated observations of individually marked, known-age Greenland white-fronted geese (*Anser albifrons flavirostris*) to determine whether cohort effects may be attributable to hatch year conditions, conditions experienced during adulthood in the years prior to successful reproduction and/or those experienced in the year of successful reproduction using commonly measured life history traits, including age at first successful reproduction (hereafter AFSR), the size of the first successful brood (SFSB) and the proportion of successful breeders by cohort (PSBC). We define 'cohort' as a group of hatch year birds marked during winter in a given year. We include environmental variables to reflect conditions experienced at each life stage. Greenland white-fronted geese are an ideal study species because they are relatively long lived (>15 years; *Weegman et al., 2016*) and encounter a variety of seasonal conditions throughout the year as they breed in west Greenland, stage during autumn and spring in Iceland and winter in Great Britain and Ireland (*Fox, Glahder & Walsh, 2003*; *Fox et al., 2014*).

## MATERIALS & METHODS

### Study area and population

From 1983 to 2003, 736 first-winter Greenland white-fronted geese were caught at Wexford Slobs (52°22′N, 6°24′W) under a ringing license (number A3136) granted to AJ Walsh from the British Trust for Ornithology. We truncated the dataset after the 2003 cohort (i.e., hatch year) to ensure adequate capture histories (i.e., compiled up to 2009) for later cohorts. Geese were caught using standard cannon-netting techniques throughout winter on baited sites and individually marked with a metal leg band, white plastic leg band and an orange neck collar (both bearing the same unique alphanumeric code; see *Warren et al., 1992*), which complied with the requirements of the National Parks and Wildlife Service (Ireland). Collar codes were legible with a 20–60× spotting scope at up to 800 m distance. Individual geese were aged (juvenile or adult) by plumage characteristics (presence/absence of white frons on face and black belly bars) and sexed by cloacal examination (*Warren et al., 1992*; *Cramp & Simmons, 1977*). AJ Walsh resighted geese weekly at Wexford throughout all winters, beginning when birds arrived in autumn. Importantly, we based all metrics of reproduction on resightings of marked Greenland white-fronted geese during winter at Wexford. Therefore, our estimates of the SFSB, AFSR and PSBC are contingent on juveniles surviving as goslings, fledging and migrating successfully to the winter quarters.

### Determining the size of the first successful brood

We determined the size of a successful brood when focal neck collared birds (with or without a mate) were observed repeatedly (>2 times) with juveniles during early winter (October–December) at Wexford. In rare cases where brood sizes differed within a winter, we used the mode. We used the SFSB instead of mean brood because very few birds were classified as having bred successfully (i.e., observed on wintering areas with young) more than once.

### Determining age at first successful reproduction and the proportion of successful breeders by cohort

We determined AFSR as the age at which a known-age individual was first repeatedly observed (>2 times within a winter at Wexford) as an adult, independent from its parents and with at least one juvenile. We analyzed AFSR individually to investigate the relative contributions of hatch year conditions, those experienced from adulthood prior to successful reproduction and those experienced in the year of successful reproduction. We calculated the PSBC as the number of birds observed with broods in a particular cohort divided by the total number of birds in that cohort. The number of birds marked in each cohort varied from 72 birds in the 1985 cohort to 9 birds in the 2001 cohort (Fig. 1A).

### Environmental metrics

We obtained North Atlantic Oscillation (NAO) data from the Climate Prediction Centre (www.cpc.ncep.noaa.gov). The NAO is a cyclical weather phenomenon that is described by pressure differences between the Azores and Iceland (*Ottersen et al., 2001*). Positive NAO phases indicate low pressure over Iceland and increased frequency of severe storms crossing the North Atlantic between Iceland and Scandinavia (*Hurrell, 1995*), whilst negative NAO phases indicate the opposite effect (i.e., high pressure and weaker storm systems). However, in west Greenland, positive NAO phases are typified by colder conditions and less precipitation, whereas negative phases are characterized by warmer conditions and more precipitation (*Stenseth et al., 2003*). We used mean NAO indices for May and December (Figs. 1B and 1C) as proxies for environmental conditions at key points in the annual cycle (i.e., for pre-nesting foraging/nesting conditions and those encountered during winter in Great Britain and Ireland), which we predicted would exert the greatest influence on fitness proxies across cohorts of Greenland white-fronted geese. In North America, positive spring and summer NAO indices have been correlated with declines in reproductive output of Arctic-nesting greater snow geese (*Chen caerulescens atlanticus*; *Morrissette et al., 2010*) and light-bellied brent geese (*B. bernicla hrota*; *Harrison et al., 2013*); however, in Greenland white-fronted geese, we would expect that positive May NAO indices would result in favorable breeding conditions (i.e., cold and dry) in west Greenland. December NAO indices may predict reproductive output during the following summer because environmental conditions during winter have been shown to contribute to explaining arrival date on breeding areas (*Saino et al., 2004a*), and breeding probability (*Sedinger et al., 2008*; *Sedinger et al., 2011*) and success (*Saino et al., 2004b*) in birds. Therefore, we would expect that cohorts which experience more positive May NAO indices in Greenland (i.e., colder and drier conditions) and negative December NAO indices on wintering areas (i.e., less storms) during their potential breeding lives would first reproduce successfully at younger ages, have larger broods and more successful breeders per cohort.

To determine whether environmental conditions experienced by cohorts through adulthood and prior to successful reproduction explained variation in AFSR and SFSB, we developed a 'breeding conditions index' (BCI) and used annual May NAO indices as a proxy for environmental conditions experienced during the breeding season (see Environmental metrics for description of May NAO). We calculated the mean May NAO

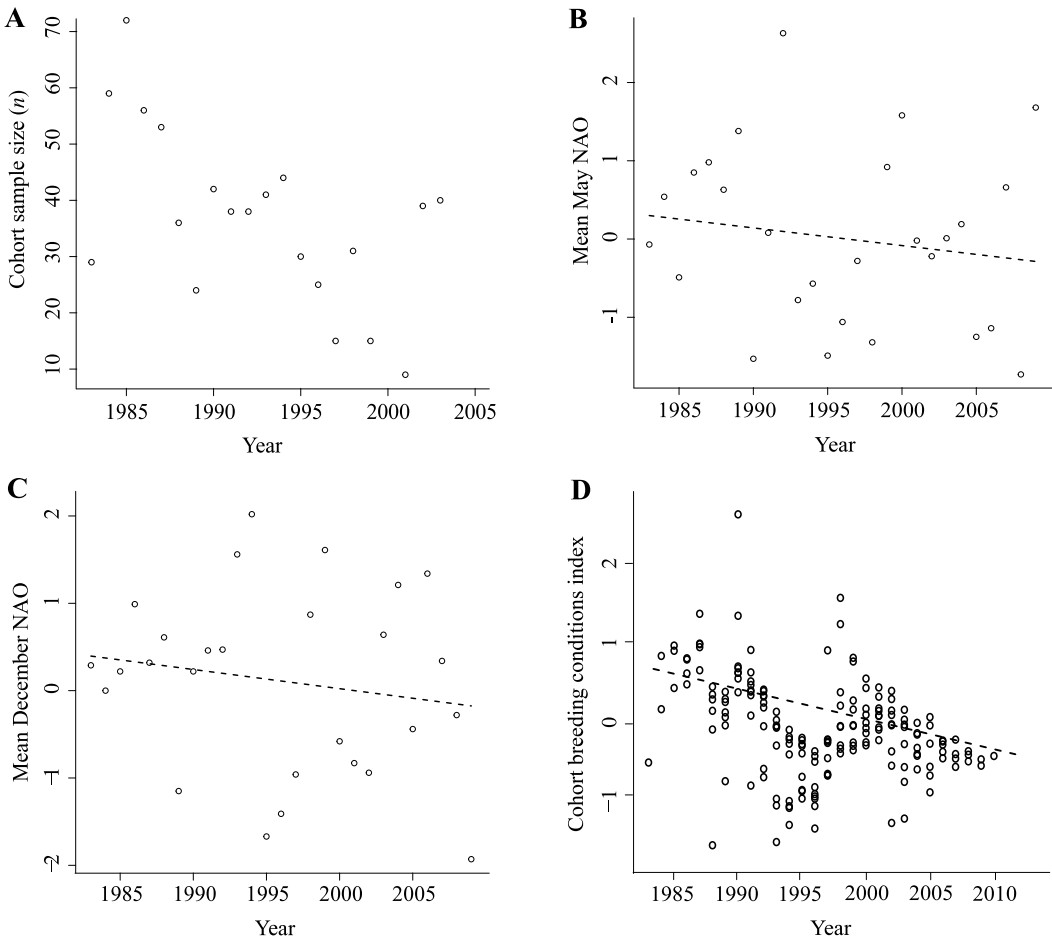

**Figure 1** **Environmental variables used in analyses of age at first successful reproduction, size of the first successful brood and the proportion of successful breeders by cohort.** (A) Greenland white-fronted goose cohort sample size (*n*), 1983–2003. All geese were marked in their first winter (i.e., were known age). Mean annual monthly (B) May NAO and (C) December NAO. These parameters were included in generalized linear mixed models of age at first successful reproduction and size of the first successful brood. (D) Cohort breeding conditions index (CBCI) by year, 1983–2003. The CBCI was calculated using May NAO indices and based on a rolling average for the years from reproductive maturity (age 2) to age 10 for each cohort, and was included in a generalized linear mixed model of the proportion of successful breeders by cohort. Positive CBCI values indicated 'good' environmental conditions across the reproductive lifetime of a cohort, whilst negative CBCI values indicated the opposite effect (i.e., poor environmental conditions). For all plots, lines were fitted using regression models with linear and quadratic terms.

indices to which individuals were exposed from age 2 (i.e., reproductive maturity) to one year prior to successful reproduction. The BCI aims to describe the cumulative conditions during the sequence of annually variable breeding opportunities to which each individual was exposed. Increasingly positive BCI scores imply exposure to a series of years with more favorable breeding conditions and increasingly negative BCI scores indicate more years of adverse breeding conditions. By using May NAO for calculation of the BCI and as a proxy for hatch and breeding year conditions, we were able to examine the influence of such conditions during each life stage on variation in AFSR and SFSB among individuals.

To investigate whether variation in the PSBC was explained by the set of environmental conditions each cohort experienced over its potential reproductive lifetime, we developed a similar index, the 'cohort breeding conditions index' (CBCI). We calculated CBCI scores using May NAO indices based on a rolling average for the years from reproductive maturity (age 2) to age 10 for each cohort (e.g., the CBCI for an age 5 bird was the average of May NAO indices at ages 2–5). The CBCI scores declined from the 1983 cohort to the 2003 cohort (Fig. 1D). An important difference between the BCI and CBCI is that the CBCI is calculated through age 10 for all cohorts (i.e., it is not truncated by successful reproduction) and is therefore a proxy for the overall set of environmental conditions to which each cohort was exposed throughout adulthood, where mean positive CBCI scores indicate 'better' breeding conditions across a cohort's adult life. By including the CBCI in analyses of the PSBC, we are able to better understand the environmental patterns influencing 'successful' and 'unsuccessful' cohorts, namely whether 'poor' conditions prevailing throughout the reproductive life of a cohort resulted in fewer successful breeders. We limited the CBCI to age 10 because incubation and brood-rearing success in geese significantly decreases beyond this age (Rockwell et al., 1993).

## Statistical analyses

We performed all analyses examining variation in the AFSR, SFSB and PSBC in Program R, version 2.14.2 (R Development Core Team, 2012). We assessed multicollinearity among variables by calculating variance inflation factors using the HH package (Heiberger, 2016); no variables had scores >3, hence, multicollinearity was minimal (see Cade, 2015).

To determine the relative contributions of hatch year conditions, those experienced during adulthood prior to the year of successful reproduction and those experienced in the year of successful reproduction on AFSR and SFSB, we fitted generalized linear mixed models with Poisson error distributions and log link functions using the lme4 package (Bates et al., 2014) and included year of successful reproduction as a random intercept (i.e., to account for unexplained variation between cohorts) and hatch year May NAO, May NAO in the year of successful reproduction, December NAO in the winter prior to successful reproduction and the BCI (i.e., average May NAO from adulthood prior to the year of successful reproduction) as fixed effects in models of AFSR and SFSB (Table 1). We included logical (i.e., interpretable) two-way interactions in the models. After initial models of AFSR and SFSB, we removed the random effect of year at first successful reproduction because it explained zero variance. We completed further analyses of fixed effects using generalized linear models.

For each potential year of successful reproduction, the response for models of the PSBC was '1' or '0' dependent on whether any bird from that particular cohort successfully bred in that year. We included cohort size ($n$) as an offset in all models to reduce bias towards larger cohorts. We fitted generalized linear mixed models using a logit link function and binomial error distribution and included cohort (i.e., hatch year) and potential year of successful reproduction (i.e., for ages 2–10) as random intercepts, and May NAO during the year of successful reproduction, December NAO in the winter prior to successful reproduction, cohort size and CBCI as fixed effects. We fitted cohort size as an interaction
**Table 1  Model structure to explain variation in age at first successful reproduction (AFSR), size of the first successful brood (SFSB) and proportion of successful breeders by cohort (PSBC).** Model structure to examine whether variation in age at first successful reproduction (AFSR), size of the first successful brood (SFSB) and proportion of successful breeders by cohort (PSBC) was due to hatch year (HY) effects, conditions prior to successful reproduction (breeding conditions index; BCI) or those experienced in the year of successful reproduction (BY).

| Response | Fixed effects (continuous covariates) | Random effects |
|---|---|---|
| AFSR | HY May NAO | Year at first successful reproduction |
| | BCI | |
| | BY December NAO | |
| | BY May NAO | |
| SFSB | HY May NAO | Year at first successful reproduction |
| | BCI | |
| | BY December NAO | |
| | BY May NAO | |
| PSBC | CBCI | Cohort |
| | BY December NAO | Year |
| | BY May NAO | |
| | Cohort $n$ | |

with all explanatory variables to account for the relationship between the ratio of successful breeders and cohort size. We also fitted other logical two-way interactions in models.

We selected top models using Akaike's information criterion (AICc; through the MuMIn package in R; *Barton, 2013*), corrected for small sample sizes ($\Delta$AICc < 6; *Burnham & Anderson, 2002*; *Richards, 2008*), and standardized coefficients by their partial standard deviations (to ensure a common denominator in model selection and account for any small collinearity between variables in each model; *Barton, 2013*; *Cade, 2015*). We calculated model-averaged coefficients for the revised model set. We applied the nesting rule (i.e., removed complex models with greater AICc values, in favor of simpler models that shared one or more of the same terms) to the top model set, eliminating so-called 'uninformative parameters' (*Arnold, 2010*). To examine model fit, we calculated Nagelkerke $R^2$ values for retained models in the top set (or the full model containing all fixed effects if models did not differ from the null; *Nagelkerke, 1991*). The relative importance of each variable was calculated as the ratio of the model-averaged coefficient divided by its standard error (*Cade, 2015*).

## RESULTS

From the cohorts hatched between 1983 and 2003, 59 (8%) of 736 marked Greenland white-fronted geese reproduced successfully only once (i.e., were repeatedly observed within a winter with young at Wexford). Just 13 birds successfully reproduced twice and two birds successfully reproduced thrice (Fig. 2). No juvenile geese were marked in 2000; thus, this cohort could not be included in the analysis. No geese from the 1996 (cohort $n = 25$), 1999 ($n = 15$) and 2003 ($n = 40$) cohorts ever reproduced successfully (i.e., no birds

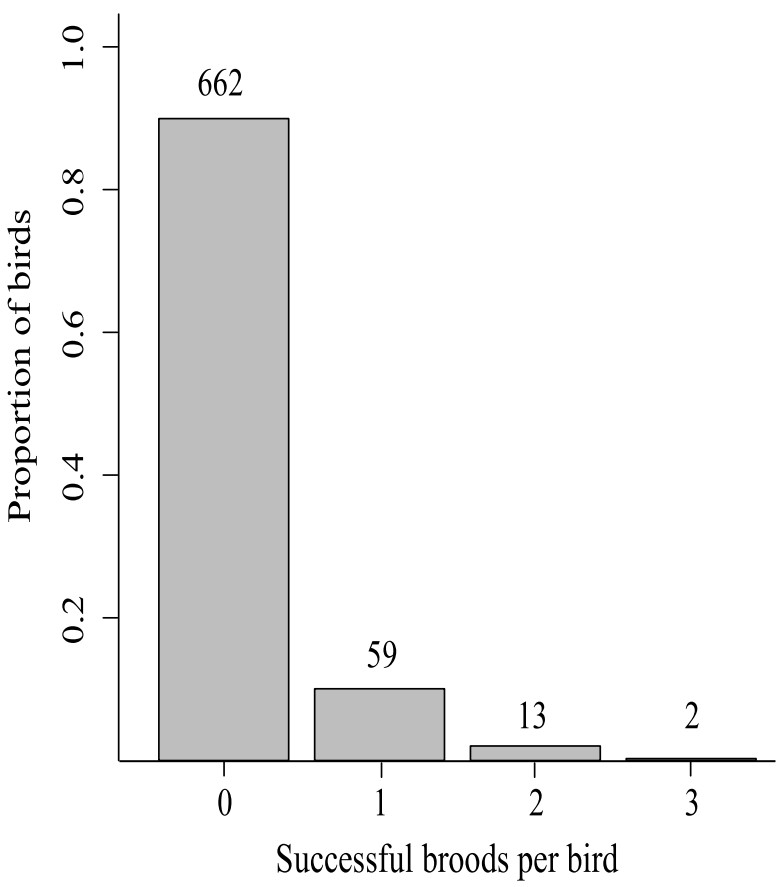

**Figure 2** The proportion of successful broods per bird (*n* subset indicated above bars) produced in the lifetimes of 736 Greenland white-fronted geese marked as first year birds at Wexford, Ireland, 1983–2003.

from these cohorts were observed repeatedly with young during winter at Wexford). Among cohorts with birds that successfully reproduced, cohort size varied from 9 birds (2001 cohort) to 72 birds (1985 cohort).

### Size of the first successful brood

Modal SFSB among ages at first successful reproduction ranged from 2 at ages 6 (min. = 1, max. = 2; *n* broods = 3), 7 (min. = 2, max. = 4; *n* broods = 6) and 10 (min. and max. = 2; *n* broods = 2) to 4 at ages 2 (min. = 1, max. = 5; *n* broods = 9) and 8 (mini. = 2, max. = 4; *n* broods = 4). Among cohorts, modal SFSB was greatest (modal brood size = 4) in the 1985 (min. = 3, max. = 6; *n* broods = 9), 1987 (min. = 1, max. = 4; *n* broods = 3) and 1988 (min. and max. = 4; *n* broods = 2) cohorts and smallest (modal brood size = 1) in the 1986 cohort (min. = 1, max. = 4; *n* broods = 8). The 'full' model (i.e., incorporating all fixed effects; Table 1) explaining variation in SFSB did not differ from the null (Nagelkerke $R^2 = 0.03$), indicating that the fixed effects we examined did not explain a significant amount of the among-individual variation.

## Age at first successful reproduction

Mean AFSR among cohorts ranged from 2-years-old (1988 cohort $n$ breeders = 2) to 8-years-old (1990 cohort $n$ breeders = 5; Fig. S1). Using the nesting rule, we retained four models from the top model set ($\Delta$AICc < 6) that included hatch year May NAO, the BCI and a two-way interaction between the BCI and May NAO in the year of successful reproduction (Nagelkerke $R^2$ estimate of top retained model = 0.25; Table 2), but did not include December NAO prior to successful reproduction. Model-averaged estimates for the standardized coefficients are presented in Table 3. The BCI was the only effect to occur in all four retained models, and its relative importance (3.49; Table 3) was three times that of hatch year May NAO and the two way interaction between the BCI and May NAO in the year of successful reproduction (relative importance = 1.22 and 1.16, respectively). Thus, variation in AFSR was explained primarily by conditions that birds experienced from adulthood prior to the year of successful reproduction, whereby birds that experienced 'good' conditions from adulthood prior to the year of successful reproduction reproduced at youngest ages (i.e., age 4; Fig. 3), but those that experienced 'poor' conditions in adulthood reproduced at oldest ages (i.e., age 7) among the birds in this study.

Variation in AFSR was also explained by a weak relationship with hatch year environmental conditions, where birds that hatched in years with 'good' conditions (i.e., positive May NAO) successfully reproduced at earlier ages (Fig. S2A), although the effect size (standardized coefficient $-0.07$, 95% confidence interval (CI) $-0.20, 0.01$) was marginally significant and less than that of the BCI (coefficient $-0.19$, 95% CI $[-0.29, -0.08]$). Similarly, variation in AFSR was explained by a weak two-way interaction between the BCI and conditions experienced in the year of successful reproduction (coefficient $-0.09$, 95% CI $[-0.24, -0.03]$), where conditions experienced in the year of successful reproduction determined the importance of conditions experienced previously in adulthood (Fig. S2B). Birds that experienced 'good' conditions in the year of successful reproduction and 'good' conditions from adulthood prior to the year of successful reproduction successfully bred at youngest ages (i.e., age 3; Fig. S2B). Birds that experienced 'good' conditions in the year of successful reproduction and 'poor' conditions in adulthood successfully bred at oldest ages (i.e., age 9). Nonetheless, we caution interpretation of results for hatch year conditions and the two-way interaction between the BCI and conditions in the year of the successful reproduction because coefficient estimates were very small (and their 95% CIs were near zero).

## Proportion of successful breeders by cohort

The proportion of each cohort that successfully returned to Wexford with young (i.e., were considered successful breeders) was greatest in the 2001 cohort (22% successful breeders) and least in the 1996, 1999 and 2003 cohorts (no successful breeders; Fig. S3). The variance explained by the cohort random intercept (0.14, standard deviation (SD) 0.38) was small and less than that of the potential breeding year (2.22, SD 1.49). Our full model explaining variation in the PSBC did not differ from the null (Nagelkerke $R^2 = 0.11$); thus we could not explain whether variation in hatch year conditions, those experienced from adulthood prior to successful reproduction or those experienced in the year of successful reproduction influenced the PSBC.

**Table 2  Top model set (ΔAICc < 6) explaining variation in age at first successful reproduction across cohorts 1983–2003 among Greenland white-fronted geese.** After the nesting rule was applied (*Richards, 2008*), we retained four models (indicated by a '√').

| | BCI[a] | BY D NAO[b] | HY M NAO[c] | BY M NAO[d] | BCI*HY M NAO | BCI*BY M NAO | HY M NAO*BY M NAO | df | logLik | AICc | ΔAICc < 6 | R[e] | Nagelkerke[f] |
|---|---|---|---|---|---|---|---|---|---|---|---|---|---|
| m1 | + | | + | + | | + | | 5 | −159.01 | 328.90 | 0.00 | √ | 0.25 |
| m2 | + | | | + | | + | | 4 | −160.77 | 330.12 | 1.22 | √ | 0.22 |
| m3 | + | | + | | | | | 3 | −161.97 | 330.28 | 1.39 | √ | 0.19 |
| m4 | + | | + | + | + | + | | 6 | −158.64 | 330.54 | 1.64 | | |
| m5 | + | | + | + | | + | + | 6 | −158.67 | 330.60 | 1.70 | | |
| m6 | + | + | + | + | | + | | 6 | −158.83 | 330.92 | 2.02 | | |
| m7 | + | | | | | | | 2 | −163.79 | 331.75 | 2.85 | √ | 0.15 |
| m8 | + | | + | | + | | | 4 | −161.67 | 331.91 | 3.01 | | |
| m9 | + | | + | + | + | + | + | 7 | −158.29 | 332.28 | 3.38 | | |
| m10 | + | | + | | + | | + | 5 | −160.76 | 332.41 | 3.51 | | |
| m11 | + | | + | + | | | | 4 | −161.92 | 332.41 | 3.52 | | |
| m12 | + | + | + | | | | | 4 | −161.97 | 332.52 | 3.62 | | |
| m13 | + | + | + | + | | + | + | 7 | −158.49 | 332.68 | 3.78 | | |
| m14 | + | + | + | + | + | + | | 7 | −158.50 | 332.69 | 3.79 | | |
| m15 | + | + | | | | | | 3 | −163.66 | 333.66 | 4.77 | | |
| m16 | + | | | + | | | | 3 | −163.68 | 333.70 | 4.80 | | |
| m17 | + | | + | + | + | | | 5 | −161.61 | 334.11 | 5.21 | | |
| m18 | + | + | + | | + | | | 5 | −161.66 | 334.21 | 5.31 | | |
| m19 | + | | + | + | | | + | 5 | −161.72 | 334.32 | 5.43 | | |
| m20 | + | + | + | + | + | + | + | 8 | −158.14 | 334.50 | 5.61 | | |
| m21 | + | + | + | + | | | | 5 | −161.92 | 334.72 | 5.82 | | |

**Notes.**
[a] Breeding conditions index (BCI).
[b] December NAO prior to successful reproduction.
[c] Hatch year (HY) May NAO.
[d] Breeding year (BY) May NAO.
[e] Retained model after application of the nesting rule.
[f] Nagelkerke R² value.

# DISCUSSION

Life histories of known-age individually marked Greenland white-fronted geese showed that only 10% of these individuals ever reproduced successfully and very few (2%) reproduced successfully more than once during their lifetime. Variation in AFSR was explained primarily by environmental conditions experienced during adulthood prior the year of successful reproduction. When birds experienced good conditions (i.e., cool and dry, positive May NAO) in adulthood prior to the year of successful reproduction, they reproduced at youngest ages. When birds experienced poor conditions (i.e., warm and wet, negative May NAO) in adulthood prior to the year of successful reproduction they reproduced at oldest ages. The influence of conditions in the year of successful reproduction on breeding success of Arctic-nesting birds has been documented in other populations (e.g., in lesser snow geese (*C. caerulescens caerulescens*) and Atlantic brant (*B. bernicla*

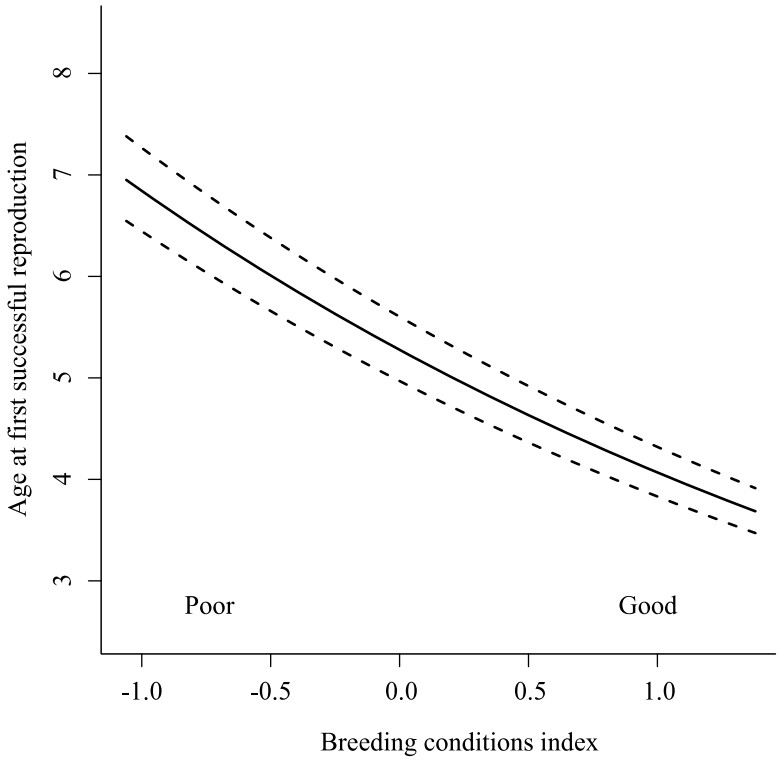

**Figure 3** **Age at first successful reproduction among Greenland white-fronted geese (1983–2003) as a function of the breeding conditions index (BCI).** The BCI was based on averaged May NAO indices from the time birds reached reproductive maturity (age 2) through one year prior to successful reproduction, where positive BCI values indicated 'good' conditions and negative values 'poor' conditions.

**Table 3** **Model-averaged estimate, 95% confidence intervals (CI) and relative importance for fixed effects in the top model set explaining variation in age at first successful reproduction in Greenland white-fronted goose cohorts 1983–2003.**

| Fixed effects | Estimate | 95% CI | Relative importance |
|---|---|---|---|
| (Intercept) | 0.00 | 0.00 | – |
| BCI[a] | −0.19 | −0.29, −0.08 | 3.49 |
| HY M NAO[b] | −0.07 | −0.20, 0.01 | 1.22 |
| BCI*BY M NAO | −0.09 | −0.24, −0.03 | 1.16 |
| BCI*HY M NAO | −0.01 | −0.15, 0.06 | 0.32 |
| HY M NAO*BY M NAO | 0.01 | −0.06, 0.15 | 0.26 |
| BY D NAO[c] | 0.004 | −0.09, 0.12 | 0.14 |
| BY M NAO[d] | 0.003 | −0.10, 0.11 | 0.07 |

**Notes.**
[a] Breeding conditions index (BCI).
[b] Hatch year (HY) May NAO.
[c] December NAO prior to successful reproduction.
[d] Breeding year (BY) May NAO.

*hrota*); *Davies & Cooke, 1983*; *Skinner et al., 1998*; *Barry, 1962*) and in this population (*Boyd & Fox, 2008*). However, we are unaware of previous studies that evaluated conditions experienced in adulthood prior to the year of successful reproduction to understand their collective impact on life history traits.

We could not explain variation in the SFSB through environmental variables measured during hatch year, adulthood prior to the year of successful reproduction or in the year of successful reproduction. Thus, we could not link conditions in adulthood prior to successful reproduction, which explained significant variation in AFSR, with those that explained variation in the SFSB. This suggests that environmental conditions influenced successful breeding, but not the size of the successful brood (i.e., birds experienced either good conditions and produced similar-sized broods, or poor conditions and did not successfully produce a brood). Alternatively, with only 74 successful breeders in 27 years in this study, it is possible that our sample was too small to explain variation in the SFSB or that factors other than those we examined here influenced the remaining variation in the SFSB.

Similarly, we could not attribute variation in the PSBC to the environmental conditions experienced from hatching to successful reproduction. However, there was large between-cohort variation in the PSBC, which suggests that we did not identify all potential sources of the variation. Individual heterogeneity could determine whether individuals from a given cohort (i.e., which experience the same environmental conditions) successfully reproduced (*Barbraud & Weimerskirch, 2005*) particularly if they experienced average or poor environmental conditions from adulthood prior to successful reproduction. For example, because of carry-over effects (*Harrison et al., 2011*), within a given cohort, high quality individuals may successfully reproduce under average environmental conditions, whereas poor quality individuals may either require multiple years of average and/or good environmental conditions to attain successful reproduction, or never successfully reproduce, despite average or good environmental conditions. In addition, analyses examining other factors at 'important' points in the annual cycle (e.g., nutrient acquisition during spring migration, which is known to influence reproductive success in migrant birds; *Weber, Ens & Houston, 1998*; *Prop, Black & Shimmings, 2003*) are needed to better understand patterns in the PSBC.

A potential source of bias in our study is the imperfect detection of individuals at their main wintering site (Wexford, Ireland), as not all collared individuals were resighted in every year of their lifetime. Thus, some birds may have successfully reproduced (i.e., returned to Wexford with young), but not been detected either individually or with their brood. To assess this bias, we filtered our data set to include only individuals that were seen in every year after marking until the end of their capture histories (i.e., those considered 'perfectly' resighted), when they either permanently emigrated or died. In total, 549 of the 736 birds (and 41 of 74 successful breeders) included this study were 'perfectly' resighted. Results from similar models of AFSR, SFSB and PSBC matched those found in analyses of all birds, specifically that the BCI explained significant variation in AFSR, where birds successfully reproduced at youngest ages when conditions in adulthood were 'good' (see Table S1). Importantly, no other effects explained variation in AFSR, and no effects explained significant variation in the SFSB or PSBC. These results suggest that any potential bias associated

with the imperfect detection of successful breeding individuals is minimal and unlikely to influence results and inferences drawn from our analyses incorporating all individuals.

Here, we have demonstrated that variation in AFSR among individual Greenland white-fronted geese was explained by the conditions birds experienced from adulthood prior to the year of successful reproduction. Since 90% of successful breeders brought young back to Wexford only once in their lifetime, these environmental conditions equate to factors influencing not only AFSR, but effectively lifetime reproductive success. Thus, the ultimate fitness of most individuals was dependent on these conditions. In this population, fitness advantages to reproducing successfully earlier seem unclear, because even those that reproduced successfully earlier in life rarely did so again (in contrast to individuals of other species that commonly successfully reproduce across multiple years; *Newton, 1989*; *Cooke, Lank & Rockwell, 1995*; *Krüger & Lindström, 2001*). However, classic life history theory predicts that for every year in which birds fail to reproduce successfully, they risk dying with zero fitness (inclusive fitness notwithstanding; *Hamilton, 1964*). Thus, individuals yet to have reproduced successfully must balance the risk of dying in the next year with reproductive attempts in variable (and increasingly suboptimal) environmental conditions. Indeed, over our study period, poor conditions became more frequent; from 1983 to 1992, a negative May NAO phase occurred in just three years, but in seven years from 1993 to 2002 and in five years from 2003 to 2012, which may explain the recent decline in productivity in this population, and the 'sink' status of the Wexford subpopulation (*Weegman et al., 2016*). Even when cohorts were exposed to good breeding year conditions, many that experienced poor conditions from adulthood reproduced later. Hence, in recent years, there have been fewer years of good breeding conditions and individuals that lived through these years were exposed to a cumulative negative effect, which caused successful reproduction at older ages. That individuals in cohorts experienced the same hatch year conditions and similar conditions from adulthood prior to successful reproduction indicates these should be studied as cohort effects. Thus, we build on previous studies which concluded that environmental conditions during hatch/birth year influenced fitness of individuals in cohorts (e.g., in red-billed choughs (*Pyrrhocorax pyrrhocorax*; *Reid et al., 2003*), greater snow geese (*Reed et al., 2003*), Soay sheep (*Forchhammer et al., 2001*) and red deer (*Cervus elaphus*; *Rose, Clutton-Brock & Guinness, 1998*)) by showing that conditions from adulthood prior to successful reproduction also influence the age at which individuals first successfully reproduce (and determine the ultimate fitness in this population).

## ACKNOWLEDGEMENTS

We thank the National Parks and Wildlife Service of Ireland, particularly the offices of J Wilson, D Norriss, O Merne and D Tierney for their support. We thank the many volunteers who have helped catch and mark Greenland white-fronted geese at Wexford over the study period, especially P O'Sullivan and the late C Wilson. We also thank D Koons, X Harrison, G Souchay, T Arnold and K Weegman for their helpful comments to earlier versions of this manuscript. Finally, we thank our respective employers for their support of this research.

### Funding

This research was funded through a joint PhD studentship from the Wildfowl & Wetlands Trust and the University of Exeter, and undertaken by MD Weegman. The funders had no role in study design, data collection and analysis, decision to publish, or preparation of the manuscript.

### Grant Disclosures

The following grant information was disclosed by the authors:
Wildfowl & Wetlands Trust.
University of Exeter.

### Competing Interests

The authors declare there are no competing interests.

### Author Contributions

- Mitch D. Weegman conceived and designed the experiments, performed the experiments, analyzed the data, contributed reagents/materials/analysis tools, wrote the paper, prepared figures and/or tables, reviewed drafts of the paper.
- Stuart Bearhop, Geoff M. Hilton and Anthony David Fox conceived and designed the experiments, performed the experiments, contributed reagents/materials/analysis tools, reviewed drafts of the paper.
- Alyn Walsh performed the experiments, contributed reagents/materials/analysis tools, reviewed drafts of the paper.

### Animal Ethics

The following information was supplied relating to ethical approvals (i.e., approving body and any reference numbers):

All marking of geese was conducted under a ringing license (number A3136) granted to AJW from the British Trust for Ornithology.

### Data Availability

All data used in these analyses are available online at FigShare.
DOI for AFSR and SFSB: 10.6084/m9.figshare.3121024.
DOI for PSBC: 10.6084/m9.figshare.3121045.

### Supplemental Information

Supplemental information for this article can be found online at http://dx.doi.org/10.7717/peerj.2044#supplemental-information.

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
