# Peer review of "Conditions during adulthood affect cohort-specific reproductive success in an Arctic-nesting goose population"

_PeerJ, doi:10.7717/peerj.2044_

## Round 0.1 · original submission · Major Revisions

All referees raise substantial concerns regarding the analytical approach and in particular the definition of age at first reproduction. This must be addressed in a revised version. You have a great case study and data set, but as for all studies, there are limitations that must be made explicit.

·

Basic reporting

The authors provided a well-written and easy to read manuscript. They well introduced the topic of their study: investigating factors influencing fitness proxies and the importance of knowing such factors.
About references, either make an alphabetic list or insert the number in the text to make it easier to find the reference.
Raw data are missing in both the main text and Supplementary Materials

Experimental design

I am one major concern about this study: the definition of age at first breeding. The authors stated that they "determined age at first breeding as the age at which a known-age individual was first repeatedly observed (>2 times within a winter at Wexford) as an adult, independent from its parents and with at least one juvenile" (Line 140-142).
Thus, the authors based their study on the fact that a breeder is an individual with a young at winter. However, I disagree with this definition. A breeder is an individual that attempts to breed. Age at first breeding is the age when an individual start to reproduce, irrespective of hatching success, fledgling success and juvenile survival. This point is very important because the authors can only observe individuals on wintering grounds, thus, their age at first breeding is the produce of an age-specific breeding probability, fledging success and juvenile survival. Juvenile survival can be very variable over time, and first migration survival is often the critical moment of first year survival (Menu et al 2005 Auk 122:479-496). Thus, this is quite unlikely that juvenile survival will not have biased their observations and their finding. Such bias are only noticed once and the authors never come back to this point afterwards, despite potential huge bias.

Aside, the sample size over time raised some questions: why such a decrease in sample size on wintering ground? Did the birds use another wintering ground in the last years?
Also, I was wondering whether observations data were available for years 2010 onward for birds marked in the 2000s. Currently, the last cohort included in the analysis is the 2003 cohort but observations stopped in 2009. Even if authors stated in a previous review that 6 years was enough based on mean age at breeding from previous years, they found an increase in the age at first breeding, thus, they should have included more years if data was available.

Validity of the findings

I have strong concerns about the analysis and the findings.
As mentioned earlier, the definition of the "age at first breeding" used here introduced potential bias. Therefore, I would like to have strong arguments showing that demographic rates such as adult and juvenile survival did not change over time, and that breeding probability is 1. Unless such statements, the finding about "age at first breeding" are not robust.
In a previous submission of this ms in another journal where I was been involved as reviewer, I followed a previous suggestion that the authors should use a capture-recapture model to estimation age-specific breeding probability to produce a better proxy of age at first breeding as it seems difficult to control for autumn survival of offspring. The suggested model was a multi-state model with both breeder and non-breeder states (the later one can also be subdivided into pre-breeder and non-breeder). The authors could take advantage of all observations they have, for both breeders and non-breeders on wintering ground and the model could allow the estimation of the transition from non-breeder to breeder states with an age effect. The inclusion of covariates is now quite easy when using MARK or E-SURGE, so they could also directly test covariates effect on breeding probability using standard procedures (e.g. Genovart et al 2013 J Anim Ecol, 82, 121-130).
In absence of other information, it seems to be the more robust way to produce inference about age at first breeding. Authors answered that there were multiple ways to answer a question. I agree on this point, however, to provide robust cues that there is no bias here is crucial.
Another concern is about their model selection process. They are using AIC to select best model which is straightforward. But, afterwards, they used a "nesting rule" proposed by Richards (2008 J Applied Ecol 45:218–227). In its paper, Richards proposed 2 alternatives ways to deal with overdispersion, either modeling the process leading to overdispersion (not possible here) or to use QAIC (AIC corrected by variance inflation factor) rather than AIC (not done here) and afterwards, to use some nesting rules. Thus, it seems that the model selection, even using the nesting rule to avoid to select too complex models may have been biased due to lack of correction for overdispersion. Furthermore, it could be interesting to provide some metrix of the variance explained by each model (simple R²) to have a measure of the relevance of the models.
Finally, the discussion is mostly based on results from the age at first breeding investigation. When looking at the top model for this variable, we can see quite easily that all parameters estimated but intercept are quite close to zero (the 95% confidence intervals should be given). Thus, the discussion is based on biological consequences of effect as they were significant although there are not. In the same manner, the authors stated that the negative slope for the effect of prior wintering census suggest a negative density-dependence effect. However, the figure 4 is lacking the 95% CI and to show the CI would have illustrated that there is no relationship between the age at first breeding and winter population size. I suggest to reduce the emphasize made based on the results and biological conclusions. Rather, it could be interesting to discuss the fact that the covariates have not effect and to discuss this point. Also, to discuss the results for the size of first brood and proportion of breeders (currently absent in the ms) would be very interesting.

Additional comments

The authors investigated a very important feature of fitness, the age at first breeding and its factors of variation. However, despite their efforts, there are several strong concerns about the methodology and lots of potential bias.
A general issue of the ms is that results and discussion are focusing on only 1 of the 3 responses variables investigated. Thus, either the ms should be restructured and to present only the result about age at first breeding or the authors should expand and discuss more the other points of their study.
A general comment on the result and the figure used is the lack of confidence intervals for the regression lines and estimates. Even if current results are "negative" ones, it is still interesting to discuss the absence of impact.

·

Basic reporting

The authors provide an excellent contribution to our understanding of cohort effects, the environmental factors that influence them, and the environmental factors that can later augment or dissipate the fitness consequences of being born in a particular environmental condition. The study is a remarkable one, and provides especially useful insight into cohort dynamics for long-lived Arctic migrants. Overall the paper is well written, and the results are interesting. However, the measurements of fitness are not defined properly, and I have some minor suggestions for some re-analyses of the data. But I stress that the paper shows real promise, and look forward to eventually reading a revised version. Below I provide more detailed comments and line edits.

Experimental design

Solid

See the General Comments below for more detail.

Validity of the findings

High Interest

See the General Comments below for more detail.

Additional comments

Main Comments

Your measures of fitness are either defined improperly, or could benefit from a more clear definition. For example, the age at first reproduction (AFR) in a bird is most commonly defined as the age at which a mating event results in at least one egg being laid (often just a day after the mating event). This is of course more easily measured for females than it is for males because of EPCs. But given your field study and methods, you are only able to measure the age at which an individual first raises (or recruits) an offspring through fledging, fall migration, and into the wintering grounds. This is often defined as a measure of reproductive success, and thus you are measuring the ‘age of first successful reproduction’ (AFSR). Many birds may breed, but then lose their nest or flightless brood, or even lose their offspring during migration. This is later eluded to, but AFR and AFSR are very different things. Your measurement of first brood size is also contingent on the definition of reproductive success. What you are really measuring is the ‘size of the first successful brood’ (which is finally mentioned on line 227, but rewording is needed throughout). Moreover, the proportion of breeders is really the ‘proportion of successful breeders’, which results from individual-level probabilities of successfully breeding. The paper needs to be rewritten with more appropriate definitions throughout.

I have a question about the sample described on line 119. Is the sample of 736 birds marked in the hatch-year a sub-sample representing only those that had perfect resighting records before departing forever (dying or permanently emigrating)? I’m wondering if there were a lot of marked HY birds, but for which a multi-state CMR analysis would be needed to analyze their encounter histories. I think the focus on the 736 is fine, but it would be good for the reader to understand the study sample, and whether a small or large fraction are perfectly fidel to the Wexford wintering area.

CBCI & lines 203-216: you evaluate this through age 10, but do you then use it to examine its effect on the proportion of breeders in a cohort on a yearly basis, or the proportion of individuals that ever bred successfully at least one in their lifetime? If the former, it would not make sense to evaluate a covariate through age 10 and then examine its effect on fitness when a cohort is only at age 5 for example (b/c the events have not yet happened to them). I would suggest that you evaluate CBCI on a sliding window basis. But you first need to clarify up front the specific measure of fitness you intend to examine with CBCI.

221-224: A less subjective cut-off for evaluating multicollinearity would be to examine the variance inflation factors (VIF), which can readily be done in R.

226-242: Is the sampling unit in the analysis the individual? As opposed to analyzing a mean across individuals in a cohort, which would not account for differing sample size among cohorts (the inappropriate ‘statistics of statistics’ approach).

244-254: Here, it would be more straightforward and appropriate to use the individual-level data: did an individual successfully breed (1) or not (0). You would retain the cohort random effect, the potential breeding year random effect, and all of the nice covariates that pertain to a cohort. Thus, you would be analyzing an individual’s probability of breeding in relation to the cohort it belongs to rather than a cohort summary of the proportion breeding.

258: Please see the new paper by Brian Cade in Ecology (2015; can’t remember if it is currently online early, or if it has page numbers). The traditional way of calculating model-averaged estimates, as outlined in the Burnham & Anderson books, can be quite flawed. Fortunately, the developers of MuMin have already implemented some routines to perform calculations using the approaches outlined in the Cade paper. But Cade himself warns that his suggested fix may not work in all situations.

263-264: Arghh! These do not measure variable importance. For a nice summary on this, see the Cade (2015) paper. Just get rid of these measurements altogether.

357-358: Or, your sample of successful breeders was too small to explain variation in size of the first successful brood size.

557-559: In the analysis did you really split the variable into categorical quartiles, or did you use the continuous covariate, and present these quartiles in the figure for visual purposes (i.e. to avoid the need for a 3D surface plot)?

Discussion: You could briefly comment on the extremes skew of individual variability in the measures of early-life reproductive success and timing, which is consistent with other long-term studies of birds and mammals. Only a few individuals contribute to the adult population, providing ample opportunity for natural selection. Moreover, reproductive success is on average low, and declining, indicative of a sink population that your team is exploring further in other papers/dissertation.

Line Comments
95-96: The transition from the bird reference to the mammal one is rather rough and the sentence does not flow well.
134: I think it might be best to delete ‘set’.
189: …environmental conditions…
204: …set of environmental conditions…
210: …environmental conditions…
230: As an example of how to more properly define the fitness metrics you are actually measuring, I would rewrite ‘and brood size’ as ‘and recruited brood size upon first successful reproduction’ or ‘and size of the recruited brood upon first successful reproduction. Earlier in the paper you would have to define what is meant by ‘recruitment’ in this particular paper (recruitment of offspring into the winter).
268: ‘once’ or ‘at least once’?
319: random cohort intercept
576: rather than ‘average model estimate’, use ‘model-averaged estimate’

Overall, this is an outstanding paper that I enjoyed reviewing very much.

Dave Koons

·

Basic reporting

No comments.

Experimental design

Methods appear appropriate to the question and well described. Research questions are clearly defined

Validity of the findings

Weegman et al present a study examining environmental effects on age at first breeding in Greenland white-fronted geese, using an impressive dataset of 21 years of ring relighting data for 736 wild-caught individuals. Using an information-theoretic approach, the authors demonstrate an interaction between breeding-year environmental conditions and conditions experienced previously in adulthood. Individuals that experienced favourable breeding year conditions in conjunction with favourable average environmental conditions prior to breeding bred at significantly younger ages than individuals that experienced the converse. The authors also demonstrate an effect of hatch year conditions on age at first breeding (i.e. silver spoon effects).

Overall I think this is an interesting paper with some important findings. The key strength of the paper is that these results demonstrate the importance of considering fitness predictors throughout an individual’s lifetime, and not solely from the year or season of breeding as many studies tend to. However I have a few concerns about methodological approaches that I feel must be dealt with before I can recommend the paper for publication. Chief among these is the manner in which the NAO indices, used as a proxy for environmental conditions, are used in the analyses. The authors appear to use the raw NAO values (and averages of raw values), rather than detrending the temporal trends of NAO and using the residuals. Failing to do so can lead to spurious associations between variables (see below), and I feel the authors must show that their results are robust to this possibility. I also list a few other minor points below.

Additional comments

Specific Comments:

Line 149: Environmental Metrics: Here you use May NAO in your analyses, but I cannot see any indication that you de-trended the NAO data and used residuals in your analyses. Harrison et al (2013) used this technique, a paper that you have cited. If there is a true association between NAO and your fitness metrics, the residuals of the relationship should be correlated independently of the raw NAO data. See Votier et al (2008) for an excellent example of the dangers of not detrending data.

Votier et al 2008; Is climate change the most likely driver of range expansion for a critically endangered top predator in northeast Atlantic waters? Biol Lett 4: 204–205. doi:10.1098/rsbl.2007.0558.

Line 253-254: What you describe here is identical to using the ‘scale’ function in R as you describe on line 238. I suggest you maintain identical terminology for both analyses. If you keep the terminology on line 253 you should define x-bar.

Line 257: Please cite Richards (2008) as justification for why one should use deltaAIC of 6 for the top model set, namely obtaining a 95% chance of retaining the model with the lowest Kullback-Leiber distance in the top model set. Note also that Richards (2008) showed that AICc provides little benefit over ‘normal’ AIC, but also that if your sample size is large enough, which in your case I suspect it is (n data points / n estimated parameters > 40), AICc converges to AIC anyway.

Line 262-263: Where you calculate variable importance as the sum of the model weights for models in which a variable appears, did you do this by re-scaling the model weights to sum to 1 in the top model set? Under the nesting rule, model weights are meaningless because you have thinned down the model set manually to exclude needlessly complex models. I would be wary of interpreting these variable importance metrics as they stand.

Line 262 r2 values: do you mean that you are calculating an average r2 value for all models in the top model set? In addition, why use Nagelkerke r2 values for mixed models and not the mixed-effects specific r2 values provided by the Nakagawa & Schielzeth method?

Line 281: seven models *that* included

Line 281-282 I see from Table 2 that you retained 7 models in total, but not all of these included the interaction you mentioned, making this sentence unclear. I would like to see statements regarding i) how many models were in the delta 6 set; ii) how many were retained and iii) how many of those contained the interaction term of interest. I think the large numbers of models in your delta 6 set suggests a great deal of uncertainty in the effects driving your observed data, so it’s important to be more transparent about how many models do or do not contain the 2 way interactions between BCI and breeding year May NAO.

Lines 309 /Brood Size Metrics: Mean and SD is often an unclear way of presenting Poisson data like counts of offspring. More useful here would be the modal brood size and perhaps minima & maxima. May I also suggest moving the brood size section up to just below the % successful breeding section? In my opinion it would fit more naturally here than after age at first breeding.

Line 371-375: If the majority of individuals only appear to breed once, what is the fitness advantage of breeding earlier? Your environmental variables don’t explain brood size, only age at first breeding and did not seem to explain breeding probability. Thus I’m not certain that with the paper framed as it currently is that your data permit you to talk about the effects of environment on lifetime breeding success. This obviously would not be the case if breeding earlier also meant increased probability of squeezing in a second successful clutch in an individual’s lifetime. I think you can strengthen this section of the discussion by making the link between environment and breeding probability more clear.

---

## Round 0.2 · Minor Revisions

The reviewers did appreciate the effort you put in revising the manuscript, and I support their assessment, the paper is much improved. One reviewer had some minor queries that need to be answered, but that should be easy and quick.

·

Basic reporting

The authors answered properly all the comments addressed by the reviewers. In particular, the redefined the main trait investigated here, age at first successful reproduction which cleared lots of previous comments. The ms is still very well-written and easy to read.

Experimental design

The study is now solid.
I only have minor comments on the methods. Based on Fig 1, it seems that some temporal trends occur in some of the covariates used (but it may just be a graphical bias). The authors did not detrend any of the covariates. It could be good to give information why they didn’t detrend the covariate.
Also, I was wondering whether a “noisy” parameter could have been introduced in the model to account for residual variance (different from the random effect already used).
Finally, why was the September NAO included in SFSB models only? This covariate could be also included in AFSR models as it can be seen as a proxy for autumn juvenile survival, and AFSR is conditional on juvenile survival.

Validity of the findings

Very interesting results.

Additional comments

I have also seen a recent work (Weegman et al. 2016 J Anim Ecol) based on the same subpopulation, and that is also quoted here. The Wexford population could be a “sink” population and some resightings of the birds were done on other wintering areas. Was information about presence of young also recorded for birds marked in Wexford? It could provide more information about breeding probability and breeding success.
Also I noticed that 75% of all birds marked were perfectly detected at Wexford but only 55% of the breeding birds. The proportion of successful breeder at Wexford seems quite constant over time. So, could it be possible that Wexford is a major wintering area for non-breeder birds or less suitable for family groups than other wintering areas?
Finally, I would suggest the authors or some of them to perform further analyses using capture-recapture models to look into age-specific breeding probability, age at first breeding. Taking advantages of sighting both at Wexford and elsewhere and of an integrated population model using counts and multi-state model (with pre-breeder, breeder and non-breeder states) could lead to estimation of juvenile survival of Greenland white-fronted geese, from breeding to wintering areas. Also, to add environmental covariates into IPM can also give additional insight into the relationship between environment and population dynamics of the species. I admit that this is speculative and concern further works conditional on the data available.

Here are minor details:
L.41 & 43, L.263: Either use digits or letter to write numbers but be consistent throughout the ms.
L.120: This may be a naïve question but what does the AJW refer to?
L.247: You are using AICc for adjustment to small sample size. Was overdispersion evaluated?

·

Basic reporting

The authors have very nicely addressed all of my previous concerns and I feel that the manuscript is much improved. They have done a very nice job of elucidating when in the life cycle experienced environmental conditions affect the "age of first successful reproduction" (observed with offspring on the wintering grounds).

Experimental design

The GLM analyses and approaches to comparing models and variables has improved. Yes, it would be nice to be able to implement a multi-state capture-mark-recapture model, but the number of events is relatively small in these datasets and such models could be too challenging to fit. Given these limitations, the authors have done a nice job of elucidating the interesting information contained in their long-term data that will be of interest.

Validity of the findings

The authors have nicely compared their findings to those based on a subset of individuals for which there are complete capture histories without imperfect detection. The results were remarkably similar between the analysis of the full dataset and the smaller subset of data. So we can be more confident that the findings are robust given the analyses that do not condition on estimation of recapture probabilities.

Additional comments

Nice Job!

·

Basic reporting

No comments

Experimental design

No comments

Validity of the findings

No comments

Additional comments

The authors have satisfactorily addressed all my concerns from the previous round of review. I have no further comments.

---

## Round 0.3 · accepted · Accept

Thanks (again) for your detailed answers to the reviewer's comments - this is a nice paper that should influence researchers working on factors influencing reproductive success and cohort effects in vertebrates.